# Galangin’s Neuroprotective Role: Targeting Oxidative Stress, Inflammation, and Apoptosis in Ischemic Stroke in a Rat Model of Permanent Middle Cerebral Artery Occlusion

**DOI:** 10.3390/ijms26051847

**Published:** 2025-02-21

**Authors:** Nut Palachai, Araya Supawat, Ratchaniporn Kongsui, Lars Klimaschewski, Jinatta Jittiwat

**Affiliations:** 1Faculty of Medicine, Mahasarakham University, Mahasarakham 44000, Thailand; nut.p@msu.ac.th (N.P.); araya.su@msu.ac.th (A.S.); 2Division of Physiology, School of Medical Sciences, University of Phayao, Phayao 56000, Thailand; ratchaniporn.ko@up.ac.th; 3Department of Anatomy, Histology and Embryology, Institute of Neuroanatomy, Medical University Innsbruck, 6020 Innsbruck, Austria; lars.klimaschewski@i-med.ac.at

**Keywords:** ischemic stroke, middle cerebral artery, galangin, anti-inflammatory, anti-apoptotic

## Abstract

The rising incidence of ischemic stroke poses significant health and healthcare burdens. Given the limitations of current therapeutic options, there is increasing interest in exploring the potential of galangin, a natural flavonoid compound, as a treatment for ischemic stroke. This study aimed to evaluate the neuroprotective effects and underlying mechanisms of galangin in mitigating oxidative stress, inflammation, and apoptosis in a rat model of permanent cerebral ischemia. Sixty male Wistar rats were divided into six groups: control; right middle cerebral artery occlusion (Rt.MCAO) with vehicle; Rt.MCAO with piracetam, a synthetic compound known as a cognitive enhancer; and Rt.MCAO with galangin administered at doses of 25, 50, and 100 mg/kg body weight. Neurological deficit scores, brain edema, neuronal density, and microglial morphology were assessed along with the activity of myeloperoxidase (MPO), a marker of inflammation, and superoxide dismutase (SOD). Additionally, the expression of key markers for inflammation and apoptosis, cyclooxygenase-2 (COX-2), interleukin-6 (IL-6), Bcl-2-associated X protein (Bax), B-cell lymphoma-extra large (Bcl-XL), and caspase-3, was analyzed to elucidate potential mechanisms. The results demonstrated that galangin treatment significantly improved neurological deficit scores, reduced brain edema, enhanced neuronal density, attenuated microglial activation, decreased MPO activity, and increased SOD activity in both the cortex and hippocampus, highlighting its neuroprotective potential. These effects were linked to the modulation of inflammatory and apoptotic pathways. Specifically, galangin significantly reduced the expression of IL-6, COX-2, Bax, and caspase-3 while increasing the levels of the anti-apoptotic protein Bcl-XL. In conclusion, galangin demonstrates significant promise as a neuroprotective agent for ischemic stroke by suppressing inflammation and apoptosis, thereby improving neurological outcomes. However, clinical trials are required to validate these preclinical findings and confirm galangin’s therapeutic efficacy in humans.

## 1. Introduction

Ischemic stroke represents a critical global health challenge, being one of the leading causes of mortality and long-term disability worldwide [1]. Its increasing prevalence along with its profound impact on healthcare costs and individual quality of life underscore the urgent need for innovative preventive and therapeutic strategies [2]. Despite advancements in medical interventions, the efficacy of current therapeutic approaches remains limited, highlighting the necessity for research-driven exploration of novel strategies for stroke management based on a deeper understanding of its pathophysiology.

The pathophysiology of ischemic stroke involves a complex cascade of events, including excitotoxicity, oxidative stress, inflammation, and apoptosis, all of which contribute to neuronal death and brain damage [3,4]. Inflammatory responses play a crucial role in the progression of cerebral ischemia, with the overproduction of inflammatory markers, such as IL-6 and COX-2, exacerbating neuronal injury [5]. Concurrently, the apoptotic pathway is activated, characterized by increased expression of pro-apoptotic proteins such as Bax and caspase-3, which leads to programmed cell death. In contrast, anti-apoptotic proteins like Bcl-XL help protect neurons by inhibiting apoptosis [6]. Additionally, MPO serves as a marker of inflammation and has been implicated in stroke development [7], while SOD is recognized as an antioxidant enzyme that protects against damage caused by free radical oxygen species (ROS) [8]. This intricate interplay between inflammation and apoptosis underscores the need for therapeutic strategies that target these multiple pathways to provide neuroprotection and improve outcomes after cerebral ischemia.

Galangin, a naturally occurring flavonoid derived from the *Alpinia officinarum* Hance plant (Zingiberaceae family), has emerged as a promising candidate for therapeutic exploration in ischemic stroke [9,10,11]. This compound has gained significant attention due to its diverse pharmacological properties, including antioxidant and anti-inflammatory effects [12,13]. These properties are particularly relevant in the context of ischemic stroke, where oxidative stress and neuroinflammation play pivotal roles in exacerbating ischemic damage.

Extensive preclinical research has provided compelling evidence of galangin’s neuroprotective potential in various neurological disorders, including ischemic stroke [11,14,15,16]. A previous publication from our lab shows that galangin exhibits substantial in vivo potential in alleviating the pathological alterations induced by cerebral ischemia, which might be attributed to its antioxidant effects and regulation of mitochondrial dynamics [16]. Its ability to scavenge ROS and modulate inflammatory pathways has been shown to confer significant protection against ischemic brain injury in experimental models [11,17]. However, despite these promising preclinical findings, the specific impact of galangin on ischemic stroke-induced brain damage remains relatively unexplored, especially in models that closely mimic the clinical scenarios observed in human stroke patients. One such model is the rat model of permanent cerebral ischemia induced by right middle cerebral artery occlusion (Rt.MCAO), which closely replicates the pathophysiological mechanisms underlying ischemic stroke in humans [18,19].

Given the paucity of data regarding the neuroprotective effects of galangin in this specific model, a significant gap exists in our understanding of its therapeutic potential in ischemic stroke management. Therefore, the primary objective of this study is to investigate the neuroprotective effects of galangin against ischemic stroke-induced brain injury in the Rt.MCAO rat model. By subjecting male Wistar rats to permanent cerebral ischemia via Rt.MCAO and administering galangin at varying doses for a specified duration (7 days), we aim to assess its efficacy in mitigating ischemic brain damage. This research assesses neurological outcomes by examining factors such as neurological deficit scores, brain edema, and neuronal density. Additionally, it seeks to explore the molecular mechanisms underlying galangin’s neuroprotective properties, with an emphasis on its effects on oxidative stress, inflammation, and apoptotic pathways. Key markers to be studied include SOD, MPO, COX-2, IL-6, Bax, Bcl-XL, and caspase-3. This study aims to determine galangin’s potential as a treatment for ischemic stroke, offering insights that could lead to new therapeutic strategies and improved outcomes for stroke patients, filling an important gap in current stroke care.

## 2. Results

### 2.1. Effect of Galangin on Neurological Deficit Score

The effect of galangin on neurological deficit score is shown in Figure 1. Rats that underwent a sham operation and received no treatment displayed no neurological deficit score. In contrast, rats subjected to Rt.MCAO and treated with a vehicle only exhibited a significant increase in neurological deficit score at both 1 and 7 days post-Rt.MCAO (*p* < 0.05 for both time points compared to the control group). Galangin treatment significantly mitigated these deficits at 1 and 7 days post-Rt.MCAO (*p* < 0.05 for both time points compared to the Rt.MCAO + vehicle group). Piracetam also significantly improved neurological deficit scores at 1 and 7 days post-Rt.MCAO (*p* < 0.05 compared to the Rt.MCAO + vehicle group). Although no significant differences were observed between the 1-day and 7-day scores for either galangin or piracetam treatments, both treatments showed a trend towards greater reduction in neurological deficit score at 7 days post-Rt.MCAO.

### 2.2. Effect of Galangin on Brain Edema

The impact of galangin on brain edema is depicted in Figure 2. The data show that rats subjected to Rt.MCAO and treated with a vehicle exhibited a significant increase in brain edema (*p* < 0.05 compared to the control group). Notably, both galangin and piracetam treatments significantly reduced brain edema (*p* < 0.05 compared to the Rt.MCAO + vehicle group).

### 2.3. Effect of Galangin on the Density of Surviving Neurons

Given the positive effects of galangin on neurological deficit score and brain edema observed in this study and considering the critical role of increased neuron density—whether through neurogenesis or reduced neurodegeneration—we also examined changes in neuron density in the cortex and hippocampal subregions, including CA1 and CA3, to explore potential underlying mechanisms. Treatment with the vehicle (DMSO) did not improve neuronal survival in these areas. In contrast, galangin (100 mg/kg BW) and piracetam (250 mg/kg BW) treatments significantly mitigated ischemic stroke-induced neuronal loss in the cortex and the hippocampus (*p* < 0.05) when compared to the Rt.MCAO + vehicle group (Figure 3).

### 2.4. Galangin Treatment Ameliorates the Alterations of Microglial Morphology Following Ischemic Stroke

The results shown in Figure 4 demonstrate the ameliorative effect of galangin on microglial morphological alterations following ischemic stroke. Microglial images were captured from the cortex (Figure 4A) and the hippocampal CA1 region (Figure 4B). In the control group, microglia in both the cortex and hippocampal CA1 region exhibited the typical ramified pattern, characterized by small somata with several primary processes that were extensively branched at secondary and tertiary levels. In contrast, microglial cells in the Rt.MCAO + vehicle group displayed an amoeboid appearance, marked by rounded cell bodies and fewer, shorter processes, which is a sign of a highly activated state, typically seen in the core of the infarcted area. However, piracetam administration mitigated ischemic stroke-induced microglial activation, as evidenced by a significant reduction in soma area and an increase in both the number and the length of branches. Notably, galangin treatment (50 and 100 mg/kg BW) remarkably attenuated microglial activation, with microglia exhibiting smaller somas and increased number and length of primary and secondary processes.

### 2.5. Effect of Galangin on Myeloperoxidase Activity

A previous study suggested that Rt.MCAO induction caused neuroinflammation and subsequent brain damage [20]. To further explore the inflammatory pathways involved, we measured MPO activity, an indicator of neutrophil infiltration and inflammation commonly associated with ischemic stroke. As shown in Figure 5, rats subjected to Rt.MCAO and treated with a vehicle exhibited a significant increase in MPO activity (*p* < 0.05 compared to the control group). Notably, treatments with both galangin and piracetam significantly reduced MPO activity compared to the Rt.MCAO + vehicle group (*p* < 0.05).

### 2.6. Effect of Galangin on Superoxide Dismutase Activity

SOD activity in the cortex and hippocampus was evaluated as well. Rats subjected to permanent Rt.MCAO exhibited a significant decrease in SOD activity compared to the control group (*p* < 0.05). However, treatment with piracetam or galangin resulted in significantly higher SOD activity in both the cortex and hippocampus compared to the Rt.MCAO + vehicle group (*p* < 0.05; Figure 6).

### 2.7. Anti-Inflammatory Effects of Galangin in Rt.MCAO-Induced Rats

Ischemic stroke reduces blood flow to the brain, depriving brain cells of oxygen and nutrients. This deprivation triggers an inflammatory response involving immune cells and cytokines. While this response is meant to aid in tissue repair, it can exacerbate brain damage by causing cellular injury and cerebral edema [21]. To evaluate the anti-inflammatory effects of galangin in Rt.MCAO-induced rats, this study analyzed COX-2 and IL-6 protein expressions in the cortex and hippocampus using Western blotting. Based on the observations that 100 mg/kg BW of galangin yielded optimal outcomes in preserving neuronal density, reducing inflammation, and modulating antioxidant enzyme activity, this dosage was chosen to examine galangin’s impact on COX-2 and IL-6 expression. The results are presented in Figure 7. Following Rt. MCAO induction, there was a significant increase in COX-2 and IL-6 expression in both brain regions (*p* < 0.05 compared to the control group). Remarkably, treatment with piracetam and galangin markedly suppressed COX-2 and IL-6 expressions in both the cortex and hippocampus (*p* < 0.05 compared to the Rt. MCAO + vehicle group).

### 2.8. Neuroprotective Role of Galangin in Rats with Rt.MCAO-Induced Injury via Activation of Anti-Apoptotic Pathways

Apoptosis, a critical process in maintaining tissue balance, plays a pivotal role in ischemic stroke, where reduced blood flow leads to neuronal deprivation of oxygen and nutrients, activating apoptotic pathways. This exacerbates neuronal death and worsens brain tissue damage [22]. Here, we assessed the effect of galangin on apoptotic markers, specifically Bax, Bcl-XL, and caspase-3, in the cerebral cortex and hippocampus. The results are presented in Figure 8. Following induction of Rt. MCAO, rats exhibited significantly increased expression of Bax and caspase-3 in both the cortex and hippocampus compared to the control group (*p* < 0.05; Figure 8C,D). Treatment with piracetam and galangin significantly attenuated these elevations compared to the Rt. MCAO + vehicle group (*p* < 0.05; Figure 8C,D). Furthermore, the impact of galangin on Bcl-XL expression was investigated, with the results shown in Figure 8B. Rats subjected to Rt. MCAO surgery showed a significant decrease in Bcl-XL expression in both brain regions (*p* < 0.05 compared to the control group). However, treatment with piracetam and galangin effectively reversed these changes (*p* < 0.05 compared to the Rt. MCAO + vehicle group).

## 3. Discussion

The pathophysiology of ischemic stroke is characterized by a multifaceted sequence of events initiated by a decline in cerebral blood flow (CBF) resulting in a lack of oxygen and nutrients for brain cells. This insufficiency triggers several harmful mechanisms, such as excitotoxicity, oxidative stress, neuroinflammation, mitochondrial dysfunction, and apoptosis, which collectively lead to neuronal death and damage to the brain tissue [23,24,25,26,27]. During ischemia, glutamate accumulates in the extracellular environment, excessively activating N-methyl-D-aspartate (NMDA) receptors. This overactivation causes an influx of calcium ions and initiates signaling cascades that contribute to neuronal injury [4]. Additionally, the ischemic condition promotes the generation of ROS, which inflicts oxidative damage on essential cellular components including lipids, proteins, and DNA [28]. This oxidative stress not only amplifies neuronal loss but also compromises the integrity of the blood–brain barrier (BBB), resulting in brain edema [29]. Therefore, the present study explored the effect of galangin on neurological deficit, brain edema, and neuronal density. Our study also confirmed that rats subjected to Rt.MCAO and treated with a vehicle only showed increased neurological deficit scores, elevated brain edema, and reduced neuronal density in the cortex and hippocampus regions CA1 and CA3 (*p* < 0.05 for all) compared to the control group. Surprisingly, galangin and piracetam treatment significantly improved neurological deficit scores already at day 1 and after 7 days post-Rt.MCAO, reduced brain edema, and mitigated neuronal loss in the cortex and hippocampus (*p* < 0.05) when compared to the Rt.MCAO + vehicle group. A previous study showed that galangin, at doses of 50 and 100 mg/kg BW, effectively alleviated neurological deficits, reduced cerebral infarct volume, and minimized cerebral edema by improving the microenvironment of the neurovascular unit at both 12 and 24 h post-MCAO [15].

MPO is an enzyme produced by activated neutrophils that plays a critical role in ischemic stroke by driving oxidative stress and inflammation [30]. MPO catalyzes the formation of ROS, leading to oxidative damage of cellular components, disruption of the BBB, brain edema, and neuronal loss [31,32]. Elevated MPO levels are associated with worse stroke outcomes, while its inhibition has been shown to reduce oxidative damage, preserve BBB integrity, and mitigate neuronal injury [7,33,34]. Moreover, increased MPO activity in ischemic stroke models correlates with the degree of neutrophil infiltration, oxidative damage, and inflammation, suggesting its pivotal role in stroke pathology [30,35,36]. This study highlights galangin’s ability to suppress MPO activity, which thereby reduces oxidative stress, neuroinflammation, and secondary damage in ischemic stroke. Our recent study demonstrated that galangin significantly decreased lipid peroxidation and increased catalase and glutathione peroxidase levels in the cortex and hippocampal regions affected by cerebral ischemia [16]. Moreover, the present study also explored the effect of galangin on SOD activity in the cortex and hippocampus. We found that galangin enhances SOD activity in the cortex and hippocampus, suggesting its antioxidant potential in mitigating oxidative stress induced by ischemic stroke similar to piracetam. The mechanism of action of galangin in the context of neuroprotection following cerebral ischemia involves key pathways and processes. Accumulating evidence indicates that galangin successfully inhibits lipid peroxidation (LPO), nitrict oxide (NO), ROS, and DNA damage [37,38]. Moreover, our earlier study demonstrated that galangin reduced LPO levels and counteracted the Rt.MCAO-induced reduction in catalase (CAT), glutathione peroxidase (GSH-Px) activities in the cortex and hippocampus [16]. A significant number of studies reported that piracetam provides neuroprotection through a complex mechanism involving several key actions. It aids in restoring membrane fluidity, enhances mitochondrial performance, and decreases oxidative stress and inflammation [39,40,41]. Furthermore, piracetam treatment markedly decreased MDA levels and enhanced the activities of CAT and glutathione (GSH) in the brains of doxorubicin-induced rats [42]. It also suppresses the activation of inflammatory pathways and limits the generation of ROS, both of which are pivotal in the progression of neurodegenerative conditions [43,44,45,46].

Following the ischemic injury, microglia are activated, and infiltrating neutrophils release pro-inflammatory mediators, including cytokines like IL-6 and enzymes such as COX-2, which drive inflammation and contribute to neuronal damage [47,48]. While initially protective, the neuroinflammatory response can become maladaptive if dysregulated, thereby exacerbating tissue damage. Ischemic conditions also trigger apoptotic pathways marked by increased expression of pro-apoptotic proteins, such as Bax and caspase-3, and reduce levels of anti-apoptotic proteins like Bcl-XL, resulting in cell death in susceptible areas like the cortex and hippocampus [19,49,50]. In contrast, anti-apoptotic proteins, like Bcl2 and Bcl-XL, play a crucial role in enhancing neuronal survival and minimizing brain infarct volume in both reperfusion and permanent occlusion stroke models [9,51,52]. This study explored the anti-inflammatory and anti-apoptotic effects of galangin in Rt.MCAO-induced rats. Our findings indicate that galangin exerts a neuroprotective effect by modulating these inflammatory and apoptotic pathways. Specifically, galangin treatment significantly suppressed COX-2 and IL-6 expression while decreasing the levels of pro-apoptotic markers (Bax and caspase-3) and upregulating the anti-apoptotic protein Bcl-XL. This suggests that galangin promotes neuronal survival by favoring anti-inflammatory and anti-apoptotic signaling, thereby reducing neuronal death. The observed decrease in Bax and caspase-3 levels indicates that galangin inhibits the mitochondrial apoptotic pathway, a key mechanism in preventing apoptosis and maintaining neuronal integrity. By enhancing Bcl-XL expression, galangin strengthens cellular defense mechanisms against ischemia-induced apoptotic stimuli, ultimately supporting cell survival and functional recovery. Growing evidence indicates that galangin, a flavonoid, has been successfully utilized in animal models of inflammation to demonstrate its anti-inflammatory properties by reducing the synthesis of IL-1, IL-6, and tumor necrosis factor-alpha (TNF-α) while enhancing the production of anti-inflammatory cytokines [9,10].

The findings from this study align with and extend previous research on the neuroprotective properties of galangin in various models of neurological injury [11,14,15]. While earlier studies have demonstrated galangin’s efficacy in reducing oxidative stress and inflammation in different contexts, the present study provides a more comprehensive evaluation of its effects in a clinically relevant model of permanent cerebral ischemia. The observed modulation of inflammatory and apoptotic pathways coupled with improvements in neuronal survival and neurological function and a reduction in brain edema, position galangin as a multifaceted therapeutic agent capable of addressing multiple aspects of ischemic stroke pathology.

Despite these promising findings, several limitations must be acknowledged. This study was conducted in an animal model, which, while providing valuable insights into potential mechanisms, may not fully replicate the complexity of human stroke pathology. Therefore, further studies are needed to confirm these results in clinical settings, including randomized controlled trials, to establish the efficacy, optimal dosing, and safety of galangin in human subjects [53]. Additionally, while this study provides evidence of galangin’s neuroprotective effects, the precise molecular targets and signaling pathways remain to be fully elucidated. Future research should incorporate in vitro models, such as oxygen–glucose deprivation (OGD) models, to further characterize galangin’s mechanism of action. Moreover, the use of pathway-specific inhibitors could help clarify the key molecular pathways involved. Immunofluorescence staining for blood–brain barrier integrity markers (zonula occludens-1 and occludin) could also determine whether galangin exerts indirect neuroprotection by preserving vascular integrity. Moreover, Iba1, a calcium-binding protein specific to microglia and macrophages, facilitates actin bundling and is widely used as a marker for microglial activation in immunohistochemistry. In stroke, Iba1 expression reflects microglial involvement in neuroinflammation, making it a valuable tool for studying post-stroke pathology and potential therapies [54,55]. Furthermore, combining TUNEL staining with NeuN and Iba1 can help distinguish neuronal apoptosis from microglial apoptosis, providing deeper insights into the interplay between inflammation and programmed cell death. While these mechanistic investigations were beyond the scope of our study, we recognize their significance and propose them as key directions for future research.

Taken all together, this study provides compelling evidence for galangin as a promising neuroprotective agent against ischemic stroke-induced brain injury. By effectively modulating inflammation and apoptotic pathways, galangin not only reduces neuronal death but also improves clinical outcomes, including neurological function and cerebral edema. These findings support the potential of galangin in stroke management, highlighting its role in addressing a critical unmet need in current therapeutic strategies. Future research should focus on translating these preclinical findings into clinical practice, aiming to develop effective treatment strategies that improve outcomes for ischemic stroke patients.

## 4. Materials and Methods

### 4.1. Test Treatments

The test compound, galangin (PubChem ID: 5281616), was sourced from Chengdu Biopurify Phytochemicals Ltd. (Chengdu, China). Piracetam, which served as the positive control [19], was provided by GlaxoSmithKline (Bangkok, Thailand) Ltd. The vehicle, DMSO, was obtained from Thermo Fisher Scientific (Waltham, MA, USA; product code: D/4121/PB15).

### 4.2. Experimental Design

A total of 60 healthy male Wistar rats (8 weeks old, weighing 250–300 g) were obtained from the Northeastern Laboratory Animal Center, Khon Kaen University, Thailand. The rats were randomly assigned to six groups (n = 10 per group) as follows:(i)Control group: The rats underwent a surgical procedure to expose the right middle cerebral artery without occlusion (sham operation) and received no treatment.(ii)Rt.MCAO + vehicle: The rats were subjected to Rt. MCAO and were administered an intraperitoneal (i.p.) vehicle (1%DMSO).(iii)Rt.MCAO + piracetam 250 mg/kg body weight (BW): The rats were subjected to Rt. MCAO and administered i.p. piracetam at a dose of 250 mg/kg BW.(iv)Rt.MCAO + galangin 25 mg/kg BW: The rats were subjected to Rt. MCAO and were administered i.p. galangin at a dose of 25 mg/kg BW.(v)Rt.MCAO + galangin 50 mg/kg BW: The rats were subjected to Rt. MCAO and were administered i.p. galangin at a dose of 50 mg/kg BW.(vi)Rt.MCAO + galangin 100 mg/kg BW: The rats were subjected to MCAO and were administered i.p. galangin at a dose of 100 mg/kg BW.

Treatments were administered via i.p. injection once daily for 7 consecutive days following Rt.MCAO induction. The rats were housed in groups of five in standard metal cages with controlled conditions: a 12 h light/dark cycle, relative humidity at ~30–60%, and temperature at 23 ± 2 °C. They had ad libitum access to water and commercial pellets.

The dosages of piracetam (250 mg/kg BW) and galangin (25, 50, and 100 mg/kg BW) were selected based on previous research, preliminary studies, and literature reviews [11,16,43]. Prior studies have determined the median lethal dose (LD50) of galangin to be >1500 mg/kg BW, indicating the non-toxic nature of the doses used in this study [56].

Cresyl violet staining and immunohistochemical staining of microglia were performed on five rats per group. The remaining 5 rats per group were used to examine molecular mechanisms that include MPO, SOD, COX-2, IL-6, Bax, Bcl-XL, and caspase-3. The sample size was calculated following a power analysis [57,58]. All tests were performed in duplicate.

### 4.3. Induction of Right Middle Cerebral Artery Occlusion

All animals were fasted overnight with free access to water before surgery. Anesthesia was induced with 5% isoflurane and maintained at 1–3% isoflurane in 100% oxygen. Focal ischemia was induced via permanent intraluminal occlusion of the right middle cerebral artery following established procedures. A 4-0 silicone-coated monofilament (USS DGTM Division of United States Surgical; Tyco Healthcare Group LP, Norwalk, CT, USA) was permanently inserted into the internal carotid artery approximately 17 mm or until slight resistance was encountered. The wound was sutured, and 10% povidone iodine solution was applied to the incision site for antiseptic care. In sham-operated animals, the arteries were exposed without monofilament insertion [44].

Humane endpoints were defined as the inability to move, wound infection post-surgery, weight loss > 20%, dehydration, dyspnea, progressive pain, lack of response to external stimuli, and bleeding from any orifice. All animals in this study survived the 8-day study period.

### 4.4. Assessment of Neurological Deficit Score

To evaluate the functional impact of cerebral ischemia, we used a 20-point modified neurological severity score to measure neurological deficits. This scoring system assesses various factors, including spontaneous activity, gait disturbances, postural signs, lateral resistance, and limb placement. Evaluations were conducted at 24 h and after 7 days following Rt.MCAO induction. The overall composite score reflects the degree of neurological impairment, with higher scores indicating more severe deficits. Detailed criteria and grading for the 20-point modified neurological severity score are outlined in Table 1 [59,60].

### 4.5. Assessment of Brain Edema

Brain edema was determined using the standard wet–dry method [59]. After the animals were sacrificed, the brains were isolated and weighed to obtain the wet weight. They were then dried in an oven (Memmert GmbH, Schwabach; Germany) at 70 °C for 24 h to measure the dry weight. The extent of brain edema was calculated using the following equation:Brain edema = ((wet weight − dry weight)/wet weight) × 100

### 4.6. Assessment of Neuronal Density

The rat brains were sectioned into serial coronal slices, 30 µm thick, covering both the cortex and the hippocampus. These sections were then stained with cresyl violet acetate solution (MilliporeSigma; Burlington, MA, USA) at 60 °C for 16 min to assess neuronal density. The cortex and hippocampus regions were examined under an Olympus light microscope (model CX23; Olympus Corporation; Tokyo, Japan) at 40× magnification. Images were taken at specific stereotaxic coordinates as outlined in a previous study [61] to measure neuronal density. Neuronal counts were performed in three consecutive fields, and the average was calculated to estimate the total neuron count per 255 μm^2^. The results were expressed as the number of neurons per 255 μm^2^.

### 4.7. Immunohistochemical Staining of Microglia

Free-floating immunoperoxidase labelling was performed on 30 µm-thick sections of the cortex and the hippocampus following previously established procedures [62]. Briefly, the sections were rinsed and incubated in 3% H_2_O_2_, followed by 3% normal horse serum (cat. no. A9647, SigmaAldrich; Merck KGaA, Darmstadt, Germany) to block nonspecific binding. The sections were then incubated overnight at 4 °C with a primary antibody solution containing mouse antiIba1 (1:200, cat. no. MABN92, SigmaAldrich; Merck KGaA). After washing the sections in 0.1 M PBS for 30 min, they were incubated for 2 h at room temperature with biotinylated donkey anti-mouse secondary antibody (1:500, cat. no. 715065150, Jackson ImmunoResearch Europe, Ltd.; Strate, UK). The sections were rinsed again in 0.1 M PBS and incubated for 1 h with extravidin peroxidase (1:1000, cat. no. E2886, SigmaAldrich; Merck KGaA) at room temperature, followed by another rinse. Immunolabeling was visualized using a nickel-enhanced 3,3′diaminobenzidine (DAB) reaction (cat. no. D12384, SigmaAldrich; Merck KGaA). Finally, the sections were washed in 0.1 M PBS, mounted on positively charged slides, dehydrated in graded alcohols, cleared in xylene, and coverslipped with mounting medium (cat. no. 107961, SigmaAldrich; Merck KGaA). To characterize morphological changes, the slides were examined at 40× magnification under a Ni-U upright microscope (Nikon Corporation; Tokyo, Japan) with NIS Element Imaging Software version 5 (Nikon Corporation) to evaluate the alterations in the cortex and hippocampus.

### 4.8. Assessment of Protein Concentration

Protein concentrations in the cortex and hippocampus were assessed using the Lowry method following published procedures [63]. Bovine serum albumin (MilliporeSigma) served as the standard for calibration.

### 4.9. Determination of MPO Activity

MPO activity was measured using the myeloperoxidase colorimetric activity assay kit (MilliporeSigma). Briefly, brain tissue samples were homogenized in four volumes of MPO assay buffer and centrifuged at 13,000× *g* for 10 min at 4 °C to remove insoluble components. The supernatant was collected, and 1–50 µL of each sample was added to a 96-well plate, adjusting the final volume of each well to 50 µL with MPO assay buffer. A master reaction mix was prepared, and 50 µL was added to each sample, blank, and positive control well. The plate was incubated in the dark at room temperature. The assay was monitored at 30, 60, and 120 min, with 2 µL of stop mix added at each interval. After a 10 min incubation to stop the reaction, 50 µL of TNB reagent was added to each well. Standard wells were prepared separately with varying concentrations of TNB reagent/standard. The plate was incubated for an additional 10 min, and absorbance was measured at 412 nm to determine MPO activity. The results were quantified by comparing absorbance values to a standard curve. Data were expressed as unit/mg protein.

### 4.10. Determination of SOD Activity

To determine SOD activity in the cortex and hippocampus using the Sigma-Aldrich kit (19160-1K-F), all reagents were prepared according to the kit instructions. The samples were diluted, and the controls were prepared as needed. In a 96-well plate, the appropriate volumes of the provided substrate were added, and reaction buffer was added to each well, followed by the test samples, standards, or controls. The reaction was initiated by adding the detection reagent and mixing gently, and the plate was incubated at 37 °C for 20 min. The absorbance was measured at 450 nm using a microplate reader. All measurements were performed in duplicate, and data were expressed as unit/mg protein.

### 4.11. Western Blotting

Upon completion of this study, the expression levels of Bax, Bcl-XL, caspase-3, COX-2, and IL-6 in the cortex and hippocampus of rats were assessed via Western blot analysis. Brain tissues were homogenized in lysis buffer (Thermo Fisher Scientific, Inc.; cat. no. 87792), and total protein concentrations were determined [64]. Equal amounts of protein (40 µg) were separated by 10% SDS-polyacrylamide gel electrophoresis, transferred to Hybond-P (PVDF) membrane (GE Healthcare Limited, Amersham Place, Little Chalfont, Buckinghamshire, UK), and incubated overnight at 4 °C with specific primary antibodies: rabbit monoclonal pro--Bax (1:500, cat. no. 14-6999-82, Thermo Fisher Scientific, Inc.), rabbit monoclonal anti-Bcl-XL (1:1000, cat. no. ab32370, Abcam; Cambridge, UK), rabbit monoclonal anti-caspase-3 (1:2000, cat. no. ab184787, Abcam), rabbit monoclonal anti-COX-2 (1:1000, ab179800, Abcam), mouse monoclonal anti-IL-6 (1:2000, ab9324, Abcam), and rabbit monoclonal anti-β-actin (1:5000, cat. no. AC026, Abclonal Biotech Co., Ltd.; Wuhan, China).

Subsequently, membranes were incubated with anti-rabbit (1:2000, cat. no. AS063, Abclonal Biotech Co., Ltd.) or anti-mouse (1:2000, cat. no. 12-349, MilliporeSigma) secondary antibodies for 1 h at room temperature. Immunoreactive proteins were visualized using chemiluminescent substrate (Supersignal West Pico; Pierce; Thermo Fisher Scientific, Inc.), and band densities were normalized to β-actin. Protein expression levels were quantified using a ChemiDocTM MP imaging system with Image Lab software (version 6.0.0 build 25, Bio-Rad Laboratories Inc.; Hercules, CA, USA).

### 4.12. Statistical Analysis

Data are presented as the mean ± standard error of the mean (SEM). Statistical significance was evaluated using one-way analysis of variance (ANOVA), followed by Tukey’s post hoc test, conducted with SPSS^®^ software (version 25, SPSS-IBM Inc.; Armonk, NY, USA). A *p*-value < 0.05 was considered statistically significant.

## 5. Conclusions

In conclusion, galangin demonstrates significant promise as a neuroprotective agent for ischemic stroke by reducing inflammation and apoptosis, thereby improving neurological outcomes. However, clinical trials are required to validate these preclinical findings and confirm galangin’s therapeutic efficacy in humans.

## Figures and Tables

**Figure 1 ijms-26-01847-f001:**
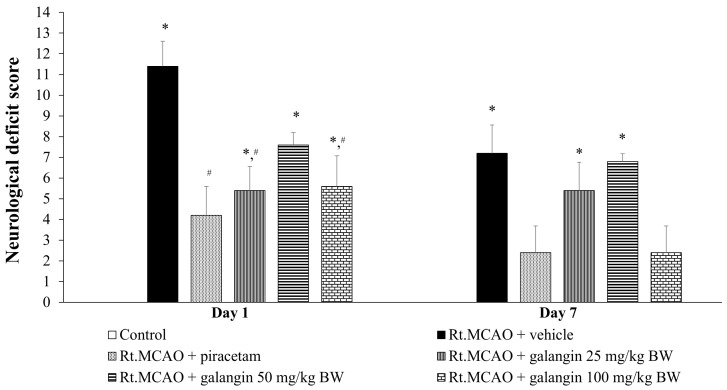
Effects of galangin on neurological deficit score. Data are expressed as the mean ± SEM (n = 5). * *p* < 0.05 when compared to the control group; # *p* < 0.05 when compared to the Rt.MCAO + vehicle group. Rt.MCAO, right middle cerebral artery occlusion; BW, body weight.

**Figure 2 ijms-26-01847-f002:**
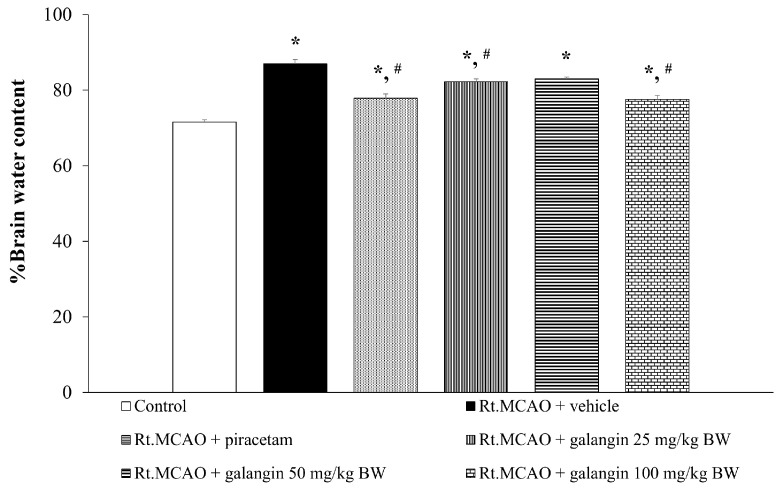
Effects of galangin on brain edema. Data are expressed as the mean ± SEM (n = 5). * *p* < 0.05 when compared to the control group; # *p* < 0.05 when compared to the Rt.MCAO + vehicle group. Rt.MCAO, right middle cerebral artery occlusion; BW, body weight.

**Figure 3 ijms-26-01847-f003:**
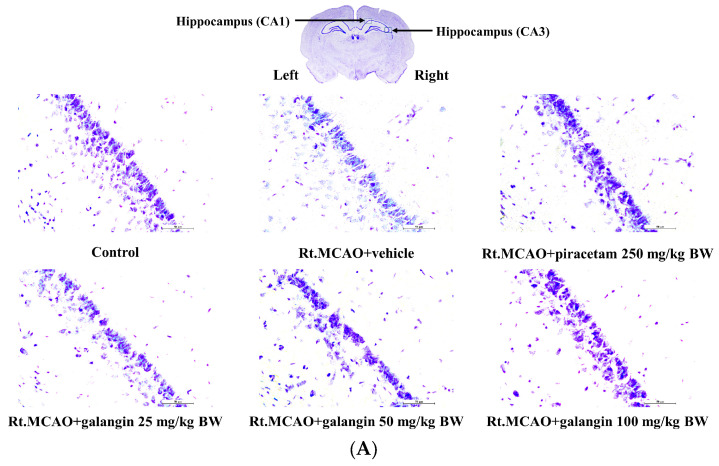
Effect of galangin on neuronal density in the cortex and hippocampus. (**A**,**B**) Representative images of Nissl-stained sections from the CA1 and CA3 regions of the rat brains, captured at 40× magnification. (**C**) A graph illustrates neuronal density in the cortex and the CA1 and CA3 regions of the hippocampus. Data are expressed as the mean ± SEM (n = 5). * *p* < 0.05 when compared to the control group; # *p* < 0.05 when compared to the Rt.MCAO + vehicle group. Rt.MCAO, right middle cerebral artery occlusion; BW, body weight.

**Figure 4 ijms-26-01847-f004:**
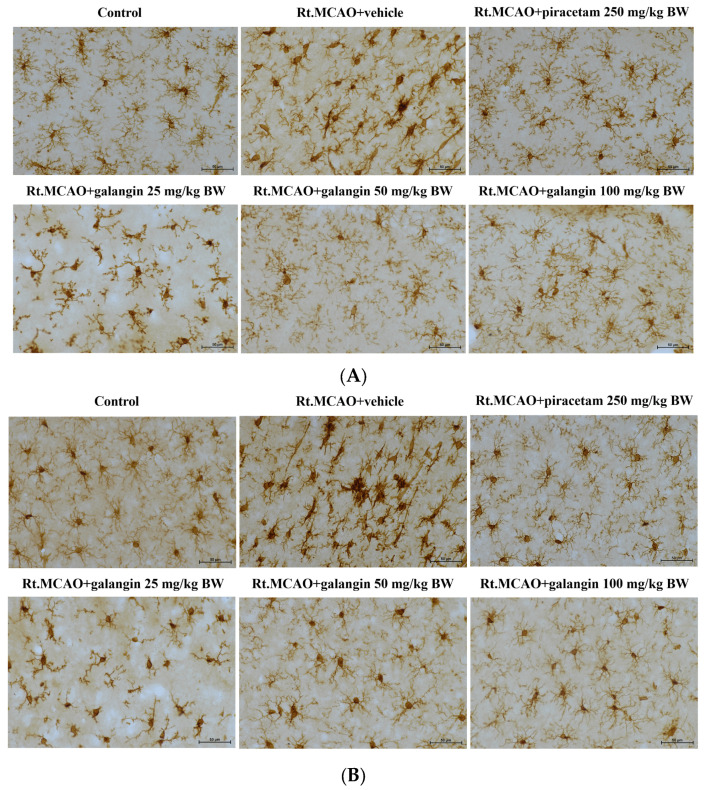
Effect of galangin on microglial activation following ischemic stroke. Iba-1 immunohistochemical staining was performed to determine microglial activation in the cortex (**A**) and the hippocampus (**B**). Representative images of ramified microglia in control group and activated microglia in the Rt.MCAO + vehicle group. Treatments with piracetam and galangin demonstrated a reduction in microglial activation. All images were taken at 40× magnification. Scale bar = 50 µm.

**Figure 5 ijms-26-01847-f005:**
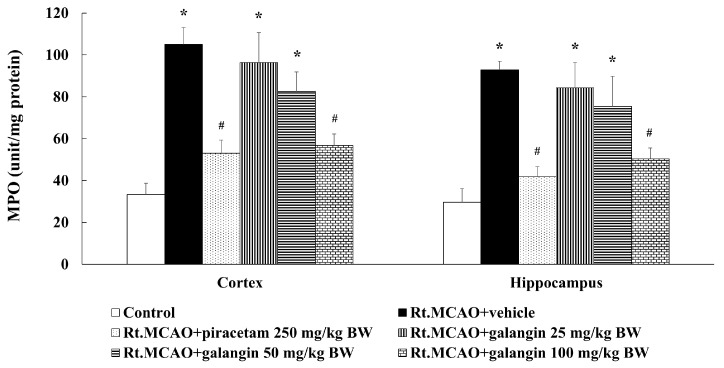
Effect of galangin on MPO activity in the cortex and hippocampus of Rt.MCAO-induced rats. Data are expressed as the mean ± SEM (n = 5). * *p* < 0.05 when compared to the control group; # *p* < 0.05 when compared to the Rt.MCAO + vehicle group. MPO, myeloperoxidase; Rt.MCAO, right middle cerebral artery occlusion; BW, body weight.

**Figure 6 ijms-26-01847-f006:**
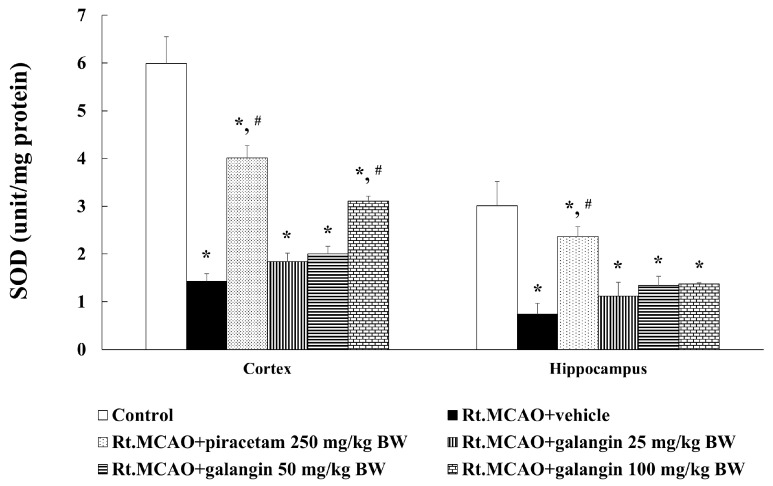
Effect of galangin on SOD activity in the cortex and hippocampus of Rt.MCAO-induced rats. Data are expressed as the mean ± SEM (n = 5). * *p* < 0.05 when compared to the control group; # *p* < 0.05 when compared to the Rt.MCAO + vehicle group. SOD, superoxide dismutase; Rt.MCAO, right middle cerebral artery occlusion; BW, body weight.

**Figure 7 ijms-26-01847-f007:**
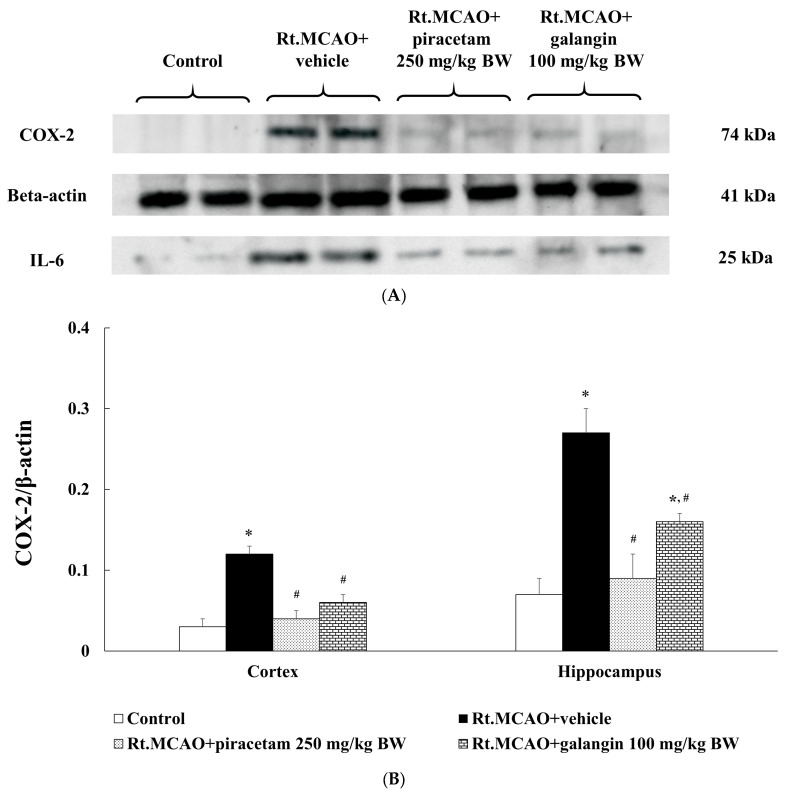
Anti-inflammatory effects of galangin in Rt.MCAO-induced rats. (**A**) Image of an immunoblot of COX-2 (74 kDa) and IL-6 (25 kDa) from the cerebral cortex. β-actin (41 kDa) was used as a loading control. (**B**,**C**) Quantitative analysis of (**B**) COX-2 and (**C**) IL-6 band density normalized with beta-actin. Data are expressed as the mean ± SEM (n = 5). * *p* < 0.05 vs. the control; # *p* < 0.05 vs. the Rt.MCAO+ vehicle group. COX-2, cyclooxygenase-2; IL-6, interleukin-6; Rt.MCAO, right middle cerebral artery occlusion; BW, body weight.

**Figure 8 ijms-26-01847-f008:**
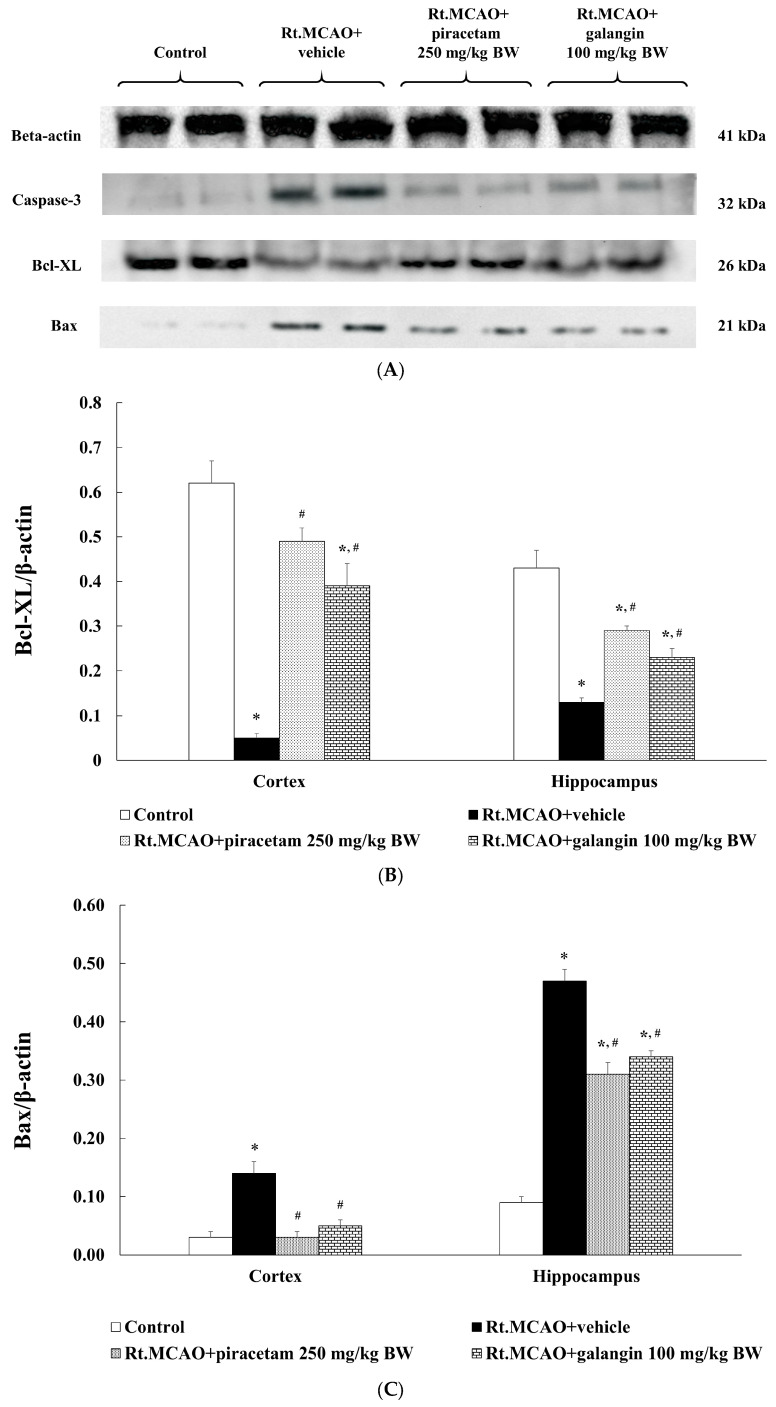
Neuroprotective role of galangin in rats with Rt.MCAO-induced injury via activation of anti-apoptotic pathways. (**A**) Image of an immunoblot of Bax (21 kDa), Bcl-xL (26 kDa), and caspase-3 (32 kDa) from the cerebral cortex. β-actin (41 kDa) was used as a loading control. (**B**–**D**) Quantitative analysis of (**B**) Bcl-xL, (**C**) Bax, and (**D**) caspase-3 band density normalized with beta-actin. Data are expressed as the mean ± SEM (n = 5). * *p* < 0.05 vs. the control; # *p* < 0.05 vs. the Rt.MCAO+ vehicle group.

**Table 1 ijms-26-01847-t001:** Criteria for assessing the neurological score using a 20-point modified neurologic severity score.

Neurological Deficit Score for Rats		
Sign	Description	Score
1. Motility, spontaneous activity	Normal	0
	Slightly reduced exploratory behavior	1
	Moving limbs without proceeding	2
	Moving only to stimuli	3
	Unresponsive to stimuli, normal muscle tone	4
	Severely reduced muscle tone, premortal signs	5
2. Gait disturbances	Straight walking	0
	Walking toward contralateral side	1
	Alternate circling and walking straight	2
	Alternate circling and walking towards paretic side	3
	Circling and/or other gait disturbances	4
	Constant circling towards paretic side	5
3. Postural signs		
3.1 Forelimb flexion	Degree of limb flexion when held by tail	0–2
3.2 Thorax twisting	Degree of body rotation when held by tail	0–2
4. Lateral resistance	Degree of resistance against lateral push	0–2
5. Limb placing		
5.1 Ipsilesional forelimb	Normal, weak, or no placing	0–2
5.2 Contralateral forelimb	Normal, weak, or no placing	0–2
Total score		20

## Data Availability

The data that support the findings of this study are available from the corresponding author upon reasonable request.

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
