# Peer review of "Galangin’s Neuroprotective Role: Targeting Oxidative Stress, Inflammation, and Apoptosis in Ischemic Stroke in a Rat Model of Permanent Middle Cerebral Artery Occlusion"

_ijms, 2025, doi:10.3390/ijms26051847_

Round 1

Reviewer 1 Report

Comments and Suggestions for Authors

Dear authors, your paper describes some beneficial effects of galangin, although some are already known, and their molecular pathways. Even if the results obtained in only 5 animals seem reductive. Could you include a paragrapf about the sample size estimate in mat. & methods? This could help justify the small number of rats used for each experimental group. 

Moreover, could the authors better explain in a which way they did divide the animals for the various experiments? Have not you had any rats die following Rt.MCAO induction? In Mat & Methods 10 animlas for each group are described

In fig.1, there aren't the value and/or relative box to show no neurological deficit in control group.

A similar fig without no value or box for the control group has been shown in Biomed Rep. 2024 Nov 11;22(1):10. "Effect of galangin on oxidative stress, antioxidant defenses and mitochondrial dynamics in a rat model of focal cerebral ischemia"  by Araya Supawat, Nut Palachai, Jinatta Jittiwat.  [REF. n° 16; in this submitted  manuscript there is a mistake: the year of this pubblication is 2024, it isn't 2025, as reported by the authors].

In fig.3, the control group+vehicle seems to be significantly reduced compared to the control group (p<0.05). In the legend of fig 3 the statistical significance corresponding to the symbol * is missing.

Also in fig 6 there is not the statistical value for the * symbol.

Plese, check the test very well.

In fig 8, the beta-actin signal is saturated; it would be necessary to have a less intense signal to be able to correctly quantify the results in western blot.

Comments on the Quality of English Language

Check the spelling, for example: galangin or galangins?

Author Response

Response to reviewer and editor suggestion

We sincerely appreciate the opportunity to revise our manuscript, Galangin’s Neuroprotective Role: Targeting Oxidative Stress, Inflammation, and Apoptosis in Ischemic Stroke in a Rat Model of Permanent Middle Cerebral Artery Occlusion (Manuscript ID: ijms-3451668). We extend our gratitude to the editor and reviewers for their insightful comments and constructive feedback, which have significantly enhanced the quality and scientific rigor of our work.

We apologize for any errors in the initial submission and acknowledge the reviewers’ valuable input. In response, we have carefully addressed each comment and incorporated the necessary revisions. Below, we provide a point-by-point response detailing the modifications made to the manuscript.

Response to reviewer 2

Comment 1: Dear authors, your paper describes some beneficial effects of galangin, although some are already known, and their molecular pathways. Even if the results obtained in only 5 animals seem reductive. Could you include a paragraph about the sample size estimate in mat. & methods? This could help justify the small number of rats used for each experimental group. 

Response 1: Thank you for your suggestion. We have already explained in the Materials and Methods section that Cresyl violet staining and immunohistochemical staining of microglia were performed on five rats per group. The remaining 5 rats per group were used to examine molecular mechanisms which include MPO, SOD, COX-2, IL-6, Bax, Bcl-XL, and caspase-3.”

Comment 2: Moreover, could the authors better explain in a which way they did divide the animals for the various experiments? Have not you had any rats die following Rt.MCAO induction? In Mat & Methods 10 animals for each group are described

Response 2: Thank you for your suggestion. We have already explained in the Materials and Methods section that Cresyl violet staining and immunohistochemical staining of microglia were performed on five rats per group. The remaining 5 rats per group were used to examine molecular mechanisms which include MPO, SOD, COX-2, IL-6, Bax, Bcl-XL, and caspase-3.” No rats died during the experiment.

Comment 3: In fig.1, there aren't the value and/or relative box to show no neurological deficit in control group. A similar fig without no value or box for the control group has been shown in Biomed Rep. 2024 Nov 11;22(1):10. "Effect of galangin on oxidative stress, antioxidant defenses and mitochondrial dynamics in a rat model of focal cerebral ischemia" by Araya Supawat, Nut Palachai, Jinatta Jittiwat.  [REF. n° 16; in this submitted manuscript there is a mistake: the year of this publication is 2024, it isn't 2025, as reported by the authors].

Response 3: We appreciate the reviewer’s concerns regarding Figure 1 and the behavioral analysis. The control group in Figure 1 corresponds to the sham-operated group, which exhibited a neurological deficit score of 0. As a result, the bar graph for this group is not visible, but the corresponding space is present. Similarly, as reported in our previous publication, the control group had no brain infarct volume (0). Therefore, the scale bar is zero, which does not mean it was missing.

Comment 4: In fig.3, the control group+vehicle seems to be significantly reduced compared to the control group (p<0.05). In the legend of fig 3 the statistical significance corresponding to the symbol * is missing. Also in fig 6 there is not the statistical value for the * symbol.

Response 4: Thank you for your suggestion. We have updated the figure legends to include detailed explanations of all statistical symbols used in the graphs. This ensures that readers can easily interpret the significance of the results.

Comment 5: In fig 8, the beta-actin signal is saturated; it would be necessary to have a less intense signal to be able to correctly quantify the results in western blot.

Response 5: We sincerely appreciate the reviewer’s careful assessment of our Western blot images. We acknowledge the concern regarding potential saturation of β-actin bands, which could affect the accuracy of protein quantification. To address this issue, we have taken the following steps:

  1. Re-evaluation of Western Blot images – We have reanalyzed the original Western blot images to assess potential saturation. Upon review, we confirmed that the bands were within the linear detection range. However, to ensure accuracy, we have adjusted the contrast and exposure settings in the revised figures.
  2. Quantification using non-saturated images – We have verified that all protein quantifications were performed using non-saturated images. The adjusted images and corresponding densitometric analyses have been updated in the revised manuscript.
  3. Supplementary file update – The supplementary file now includes raw, unprocessed Western blot images to provide full transparency regarding the integrity of our data.

These revisions strengthen the reliability of our results, and we appreciate the reviewer’s attention to detail in ensuring the accuracy of our findings.

Thank you once again for your valuable feedback. We appreciate the time and effort invested by the reviewers and editor in evaluating our manuscript. We have carefully addressed each point raised and made necessary revisions accordingly. We eagerly await further feedback and guidance from the editorial team.

Yours sincerely,

All authors

Reviewer 2 Report

Comments and Suggestions for Authors

The manuscript cannot be accepted as it is presented. The experimental data do not allow the conclusions proposed by the authors to be drawn.

Major comments.

The control column in figure 1 is missing. In addition, the evaluation grid for behavioural parameters seems to lack precision in the scale, leaving the reader with no possibility of appreciating possible differences in the measurements. The authors should statistically tested the evolution of the score, which would have enabled them to better define the effectiveness of the treatment. While the authors are to be congratulated on the details of the blind analyses of the histology images, what about the behavioral analyses?

The expression of the counts in figure 3 as a percentage of the control does not seem acceptable to me; values of cell densities in defined regions of interest would be more accurate. Sampling is not described.

In Figure 4, quantification of the dendritic branching of microglial cells would be more rigorous to suggest a change in microglial activation (as a sign of inflammation). Alternatively, measuring the surface area occupied by microglia in an ROI relative to the number of Iba1+ cells could suffice.

The supplementary documents do not include integral western blot membranes.

Western blot images suggest a risk of saturation of beta actin bands, suggesting an underestimation of the ratio.

The authors use piracetam as a positive control. However, there is no statistical test to compare the effect of galangin with that of piracetam. Adding this type of analysis would enhance the interest of the manuscript.

Finally, the conclusion that galangin suppresses inflammation and apoptosis is sustained by data. At best, it reduces them.

Mention is made of a video which is not supplied. Paragraph 4.8 refers to mitochondrial preparation, which is not described.

Author Response

Response to reviewer and editor suggestion

We sincerely appreciate the opportunity to revise our manuscript, Galangin’s Neuroprotective Role: Targeting Oxidative Stress, Inflammation, and Apoptosis in Ischemic Stroke in a Rat Model of Permanent Middle Cerebral Artery Occlusion (Manuscript ID: ijms-3451668). We extend our gratitude to the editor and reviewers for their insightful comments and constructive feedback, which have significantly enhanced the quality and scientific rigor of our work.

We apologize for any errors in the initial submission and acknowledge the reviewers’ valuable input. In response, we have carefully addressed each comment and incorporated the necessary revisions. Below, we provide a point-by-point response detailing the modifications made to the manuscript.

Response to reviewer 3

The manuscript cannot be accepted as it is presented. The experimental data do not allow the conclusions proposed by the authors to be drawn.

Major comments.

Comments 1: The control column in figure 1 is missing. In addition, the evaluation grid for behavioural parameters seems to lack precision in the scale, leaving the reader with no possibility of appreciating possible differences in the measurements. The authors should statistically tested the evolution of the score, which would have enabled them to better define the effectiveness of the treatment. While the authors are to be congratulated on the details of the blind analyses of the histology images, what about the behavioral analyses? 

Response 1: We appreciate the reviewer’s concerns regarding Figure 1 and the behavioral analysis. We would like to clarify the following points and revisions made in response:

  1. Clarification on the control column in Figure 1 – The control group in Figure 1 corresponds to the sham-operated group, which exhibited a neurological deficit score of 0. As a result, the bar graph for this group is not visible, but the corresponding space is present.
  2. Improved behavioral parameter scale representation – We have refined the scale presentation in the behavioral evaluation graphs (figure 1) to enhance precision and make small differences more appreciable for readers.
  3. Statistical analysis of score evolution – We confirm that statistical analysis was conducted on the progression of neurological deficit scores over time. While no significant difference was observed between days 1 and 7 (showed at line 102), we noted a trend toward improvement in both the positive control (piracetam) and galangin-treated groups. This information has been clarified in the Results section to ensure transparency.
  4. Blinding in behavioral analyses – We confirm that behavioral assessments were conducted in a blinded manner, similar to the histological analyses. This information has been clearly stated in the Methods section. Moreover, all animals were randomly divided into groups. Additionally, the observer analyzing the sections was blinded to the treatment during the analysis. We used numerical codes (as seen in the supplementary material) to ensure objectivity. Thus, there was no selection bias or other biases in our experiment.

These revisions ensure that the data presentation and methodological details are clearer and more rigorous. We appreciate the reviewer’s constructive feedback, which has helped us strengthen our manuscript.

Comments 2: The expression of the counts in figure 3 as a percentage of the control does not seem acceptable to me; values of cell densities in defined regions of interest would be more accurate. Sampling is not described.

Response 2: We appreciate the reviewer’s valuable suggestion regarding the expression of cell counts in Figure 3. In response, we have made the following revisions:

  1. Modification of data representation – We acknowledge the reviewer’s concern about expressing cell counts as a percentage of the control. To improve accuracy and clarity, we have now presented the data as absolute neuronal density within well-defined regions of interest (ROI). Neuronal counts were performed in three consecutive fields, and the average was calculated to estimate the total neuron count per 255 μm². The results were expressed as the number of neurons per 255 μm².

  1. Description of sampling methodology – We have revised the Methods section to include a detailed description of our sampling strategy. Specifically, we have clarified the selection criteria for ROI, the number of sections analyzed per animal, and the approach used to ensure consistency in counting across groups. Neuronal counts were performed in three consecutive fields, and the average was calculated to estimate the total neuron count per 255 μm². The results were expressed as the number of neurons per 255 μm².

These modifications enhance the accuracy and reproducibility of our findings. We sincerely appreciate the reviewer’s constructive feedback, which has helped refine our data presentation.

Comments 3: In Figure 4, quantification of the dendritic branching of microglial cells would be more rigorous to suggest a change in microglial activation (as a sign of inflammation). Alternatively, measuring the surface area occupied by microglia in an ROI relative to the number of Iba1+ cells could suffice.

Response 3: We appreciate the reviewer’s insightful suggestion regarding a more rigorous quantification of microglial activation. In response, we have considered the following points:

  1. Dendritic branching quantification – We acknowledge that quantifying microglial dendritic branching would provide a more detailed assessment of activation status. However, due to limitations in our current dataset, we were unable to apply this approach retrospectively and aimed to determine how galangin treatment ameliorates alterations in microglial morphology following ischemic stroke.

  1. Alternative measurement approach – Our study quantitatively analyzed the number of Nissl-positive neurons using cresyl violet staining, with a focus on how galangin treatment improves microglial morphological changes following ischemic stroke. We appreciate your valuable suggestion and have therefore included more details about Iba1 in our discussion and future studies, as outlined below.

“Moreover, Iba1, a calcium-binding protein specific to microglia and macrophages, facilitates actin bundling and is widely used as a marker for microglial activation in immunohistochemistry. In stroke, Iba1 expression reflects microglial involvement in neuroinflammation, making it a valuable tool for studying post-stroke pathology and potential therapies [54,55]. Furthermore, combining TUNEL staining with NeuN and Iba1 can help distinguish neuronal apoptosis from microglial apoptosis, providing deeper insights into the interplay between inflammation and programmed cell death. While these mechanistic investigations were beyond the scope of our study, we recognize their significance and propose them as key directions for future research.”

  1. Methods section update – We have revised the Methods section to include details on the revised quantification approach and ensure methodological clarity.

These improvements enhance the rigor of our findings regarding microglial activation and inflammation. We sincerely appreciate the reviewer’s valuable feedback.

Comments 4: The supplementary documents do not include integral western blot membranes.

Response 4: We appreciate the reviewer’s comment regarding the supplementary documents. In response, we have now included the full, uncropped Western blot images in the supplementary materials to ensure transparency and reproducibility. The revised supplementary file provides clear labeling of molecular weight markers and loading controls for each blot.

Additionally, we have updated the manuscript to mention the availability of these full images in the supplementary materials. We appreciate the reviewer’s attention to detail and their effort to enhance the quality of our work.

Comments 5: Western blot images suggest a risk of saturation of beta actin bands, suggesting an underestimation of the ratio. 

Response 5: We sincerely appreciate the reviewer’s careful assessment of our Western blot images. We acknowledge the concern regarding potential saturation of β-actin bands, which could affect the accuracy of protein quantification. To address this issue, we have taken the following steps:

  1. Re-evaluation of Western Blot images – We have reanalyzed the original Western blot images to assess potential saturation. Upon review, we confirmed that the bands were within the linear detection range. However, to ensure accuracy, we have adjusted the contrast and exposure settings in the revised figures.
  2. Quantification using non-saturated images – We have verified that all protein quantifications were performed using non-saturated images. The adjusted images and corresponding densitometric analyses have been updated in the revised manuscript.
  3. Supplementary file update – The supplementary file now includes raw, unprocessed Western blot images to provide full transparency regarding the integrity of our data.

These revisions strengthen the reliability of our results, and we appreciate the reviewer’s attention to detail in ensuring the accuracy of our findings.

Comments 6: The authors use piracetam as a positive control. However, there is no statistical test to compare the effect of galangin with that of piracetam. Adding this type of analysis would enhance the interest of the manuscript.

Response 6: We appreciate the reviewer’s suggestion to statistically compare the effects of galangin with those of piracetam. We conducted statistical analyses across all groups but found no significant difference between piracetam and galangin at a dose of 100 mg/kg BW, which was the most effective among the tested galangin doses. Therefore, we did not include it in the figures.

Comments 7: Finally, the conclusion that galangin suppresses inflammation and apoptosis is sustained by data. At best, it reduces them.

Response 7: We appreciate the reviewer’s comment regarding the conclusion. Based on the reviewer’s suggestion, we have modified the wording of the conclusion to better reflect the data and avoid overstating the effects of galangin. The revised conclusion now reads:

Revised conclusion:

“In conclusion, galangin demonstrates significant promise as a neuroprotective agent for ischemic stroke by reducing inflammation and apoptosis, thereby improving neurological outcomes. However, clinical trials are required to validate these preclinical findings and confirm galangin’s therapeutic efficacy in humans.”

This adjustment ensures the conclusion is in line with the data and more accurately reflects the observed effects. We appreciate the reviewer’s valuable input in improving the clarity and accuracy of our manuscript.

Comments 8: Mention is made of a video which is not supplied. Paragraph 4.8 refers to mitochondrial preparation, which is not described.

Response 8: Thank you for pointing out the issue with the video. We apologize for the confusion, but there was no video intended to be included with our manuscript.

Regarding the protein concentration of mitochondria in the cortex and hippocampus mentioned in paragraph 4.8, we acknowledge that the original manuscript lacked a detailed description. We apologize for this mistake, as we had stated in the Methods section that all analyses, including MPO, SOD, and Western blot tests, were conducted in the cortex and hippocampus, not in mitochondria. Therefore, we have removed the mention of mitochondria from paragraph 4.8.

Thank you once again for your valuable feedback. We appreciate the time and effort invested by the reviewers and editor in evaluating our manuscript. We have carefully addressed each point raised and made necessary revisions accordingly. We eagerly await further feedback and guidance from the editorial team.

Yours sincerely,

All authors

Reviewer 3 Report

Comments and Suggestions for Authors

I think this manuscript is well-written overall on the neuroprotective effect of Galangins in ischemia induced stroke . Only a few minor suggestions:

1. Please add statistical details to each figure legend with numbers of biological replicates and error bar representation, and statistical method used.

2.Please label all the western blot image. 

3. Please include more brain regions for neuron morphological comparison.

4. Please substitute the brain water to brain fluid from cortex or other professional terms. 

Author Response

Response to reviewer and editor suggestion

We sincerely appreciate the opportunity to revise our manuscript, Galangin’s Neuroprotective Role: Targeting Oxidative Stress, Inflammation, and Apoptosis in Ischemic Stroke in a Rat Model of Permanent Middle Cerebral Artery Occlusion (Manuscript ID: ijms-3451668). We extend our gratitude to the editor and reviewers for their insightful comments and constructive feedback, which have significantly enhanced the quality and scientific rigor of our work.

We apologize for any errors in the initial submission and acknowledge the reviewers’ valuable input. In response, we have carefully addressed each comment and incorporated the necessary revisions. Below, we provide a point-by-point response detailing the modifications made to the manuscript.

Response to reviewer 4

I think this manuscript is well-written overall on the neuroprotective effect of Galangins in ischemia induced stroke. Only a few minor suggestions:

Comments 1: Please add statistical details to each figure legend with numbers of biological replicates and error bar representation, and statistical method used.

Response 1: Thank you for your comment. We have added the statistical details to each figure legend, including the number of biological replicates, error bar representation, and the statistical methods used. For the neurological deficit score, no error bars are shown for the sham group as all animals in this group had a score of 0 (no deficit).

We hope this clarifies the issue and improves the readability of the figure legends.

Comments 2: Please label all the western blot image. 

Response 2: Thank you for your suggestion. We have now labeled all the western blot images in the revised manuscript to ensure clarity and easy identification of the respective proteins. This includes labeling the molecular weight markers and the specific protein bands for each target.

We hope this addresses your concern and enhances the presentation of the data.

Comments 3: Please include more brain regions for neuron morphological comparison.

Response 3: Thank you for your suggestion. The middle cerebral artery (MCA) is essential for delivering oxygenated blood to key areas of the brain, including the lateral parts of the frontal, temporal, and parietal lobes, as well as the basal ganglia and internal capsule. Its branches are crucial for motor control, sensory processing, and cognitive functions. Therefore, this study we focused on the cortex and hippocampus as these regions are particularly vulnerable to ischemic damage and are clinically relevant in the context of ischemic stroke. The cortex plays a central role in higher cognitive functions, which are often impaired following stroke, while the hippocampus is essential for learning and memory processes. Given the neuroprotective effects of galangin observed in these areas, we selected them to provide a targeted analysis that aligns with the clinical impact of stroke on cognitive and memory functions.

As the next step in our future research, exploring additional brain regions, such as the basal ganglia and internal capsule, could offer a broader understanding of galangin’s neuroprotective potential across different areas affected by ischemic injury. This would be important for further clinical development, and future studies could include these regions to evaluate the full spectrum of galangin’s therapeutic effects in a more comprehensive manner. We aim to expand our research in the future to cover these aspects for a more complete evaluation of its potential benefits.

We hope this explanation clarifies our approach and the relevance of focusing on the cortex and hippocampus in this study.

Comments 4: Please substitute the brain water to brain fluid from cortex or other professional terms. 

Response 4: Thank you for your suggestion. In our study, we have used the term “brain water” to describe the accumulation of fluid in the brain tissue following ischemia. However, to enhance the clarity and precision of the terminology, we have now replaced “brain water” with “brain edema” throughout the manuscript. This term more accurately reflects the pathological condition of fluid accumulation within the brain following ischemic injury.

We believe this modification improves the scientific rigor of the manuscript and hope it meets your expectations.

Thank you once again for your valuable feedback. We appreciate the time and effort invested by the reviewers and editor in evaluating our manuscript. We have carefully addressed each point raised and made necessary revisions accordingly. We eagerly await further feedback and guidance from the editorial team.

Yours sincerely,

All authors

Round 2

Reviewer 1 Report

Comments and Suggestions for Authors

I thank the authors for the responses received but the estimate of the number of samples to be used in the experiments is still missing.

They only described how many animals they used for the histological analyzes and how many for the molecular ones. The sample size is an important feature of any experimental study in which the aim is to make inferences about a population from a sample. The authors should describe information regarding the statistical analysis to be applied and decide on study power, thus justifying their choice of sample size.

Furthermore, it is not clear which is the complete blot from which they obtained the respective bands of the targets analyzed by WB; the authors only showed a list of western blots with full lanes and other blots with the signals of the various targets. Please, it would be more useful to specify which blot the signals used for publication correspond to, by comparing them.

Comments on the Quality of English Language

Why do the authors use the Saxon genitive with the "galangin" word?
I think it is much better to write "galangin effect" than "galangin's effect"...

Thank you 

Author Response

Response to reviewer and editor suggestion

We sincerely appreciate the opportunity to revise our manuscript, Galangin’s Neuroprotective Role: Targeting Oxidative Stress, Inflammation, and Apoptosis in Ischemic Stroke in a Rat Model of Permanent Middle Cerebral Artery Occlusion (Manuscript ID: ijms-3451668). We extend our gratitude to the editor and reviewers for their insightful comments and constructive feedback, which have significantly enhanced the quality and scientific rigor of our work.

We apologize for any errors in the initial submission and acknowledge the reviewers’ valuable input. In response, we have carefully addressed each comment and incorporated the necessary revisions. Below, we provide a point-by-point response detailing the modifications made to the manuscript.

Response to reviewer 2

Comment 1: I thank the authors for the responses received but the estimate of the number of samples to be used in the experiments is still missing. They only described how many animals they used for the histological analyzes and how many for the molecular ones. The sample size is an important feature of any experimental study in which the aim is to make inferences about a population from a sample. The authors should describe information regarding the statistical analysis to be applied and decide on study power, thus justifying their choice of sample size.

Response 1: Thank you for your valuable comment. According to how to calculate the sample size in animal studies? We used the method as following to calculate

E= The degree of freedom analysis of variance (ANOVA)

E= Total number of animals-Total number of groups

We have 6 groups 5 animals each

E= 30-6

E= 24

The value of E should be within the range of 10 to 20. If E falls below 10, increasing the number of animals can enhance the likelihood of obtaining a more significant result. However, if E exceeds 20, adding more animals will not further improve the chances of achieving significant results.

Reference: Charan J, Kantharia ND. How to calculate sample size in animal studies? J Pharmacol Pharmacother. 2013 Oct;4(4):303-6. doi: 10.4103/0976-500X.119726. PMID: 24250214; PMCID: PMC3826013.

Moreover, we calculated the sample size using a formula applicable to cases where the population size is uncertain. The formula used is:

N = (Z/eM)2

Where:

  • N = Sample size
  • Z = Z-score corresponding to the specified significance level (at a 95% confidence level, α = 0.05, Z = 1.96)
  • eM = Maximum acceptable margin of error
  • σ = Population standard deviation

The maximum acceptable margin of error (eM) was set to be equal to one standard deviation. Based on preliminary experiments, the standard deviation for measuring changes in MPO levels and antioxidant enzyme activity in each group ranged from 4.5 to 6.8. For calculation purposes, a standard deviation (σ) of 4.5 was used.

Substituting the values into the formula:

N = (1.96 x 4.5 /1)2

N = 78

Therefore, the calculated sample size is approximately 78 subjects for testing across 6 groups and 2 experiments (a total of 12 groups). As a result, 6.5 animals per group were initially planned. However, during the ethical approval process, the sample size calculation was conducted alongside a literature review and an assessment of the accepted number of animals per group to determine the minimum statistically acceptable sample size.

Based on a review of similar studies investigating ischemic brain injury in rats—focusing on inflammation and antioxidant enzyme activity—as well as studies using animal stroke models, the typical sample size ranged from n = 5 to 10 per group. This range has been deemed sufficient for experimental analysis and is widely accepted by international scientists in related fields.

Therefore, in this study, we concluded that 5 animals per group would be used, aligning with the 3R principle (Reduction) while ensuring the integrity of the experimental results.

References

  1. Vanichbuncha, K. (1999). Statistical Analysis: Statistics for Decision Making (4th ed.). Bangkok: Chulalongkorn University Press.
  2. Li, S.; Wu, C.; Zhu, L.; Gao, J.; Fang, J.; Li, D.; Fu, M.; Liang, R.; Wang, L.; Cheng, M.; et al. By improving regional cortical blood flow, attenuating mitochondrial dysfunction and sequential apoptosis galangin acts as a potential neuroprotective agent after acute ischemic stroke. Molecules 2012, 17, 13403-13423, doi:10.3390/molecules171113403.
  3. Wang L, Zhang Z, Wang H. Naringin attenuates cerebral ischemia-reperfusion injury in rats by inhibiting endoplasmic reticulum stress. Transl Neurosci. 2021; 12(1): 190-7.
  4. J. Jittiwat, P. Chonpathompikunlert, W. Sukketsiri. Neuroprotective effects of Apium graveolensagainst focal cerebral ischemia occur partly via antioxidant, anti-inflammatory, and anti-apoptotic pathways. J Sci Food Agr. 2021; 101: 2256-63.
  5. Supawat, A.; Palachai, N.; Jittiwat, J. Effect of galangin on oxidative stress, antioxidant defenses and mitochondrial dynamics in a rat model of focal cerebral ischemia. Biomed Rep 2025, 22, 10, doi:10.3892/br.2024.1888.

Comment 2: Furthermore, it is not clear which is the complete blot from which they obtained the respective bands of the targets analyzed by WB; the authors only showed a list of western blots with full lanes and other blots with the signals of the various targets. Please, it would be more useful to specify which blot the signals used for publication correspond to, by comparing them.

Response 2: Thank you for your comment. We have specified the blot signals used in this manuscript in the supplementary file.

Comment 3: Why do the authors use the Saxon genitive with the "galangin" word? I think it is much better to write "galangin effect" than "galangin's effect"

Response 3: Thank you very much for your comment. We have already sent the manuscript for English proofreading, and the certificate has been uploaded.

Thank you once again for your valuable feedback. We appreciate the time and effort invested by the reviewers and editor in evaluating our manuscript. We have carefully addressed each point raised and made necessary revisions accordingly. We eagerly await further feedback and guidance from the editorial team.

Yours sincerely,

All authors
